# MambaMOS: LiDAR-based 3D Moving Object Segmentation with Motion-aware State Space Model

## ABSTRACT

LiDAR-based Moving Object Segmentation (MOS) aims to locate and segment moving objects in point clouds of the current scan using motion information from previous scans. Despite the promising results achieved by previous MOS methods, several key issues, such as the weak coupling of temporal and spatial information, still need further study. In this paper, we propose a novel LiDAR-based 3D Moving Object Segmentation with Motion-aware State Space Model, termed MambaMOS. Firstly, we develop a novel embedding module, the Time Clue Bootstrapping Embedding (TCBE), to enhance the coupling of temporal and spatial information in point clouds and alleviate the issue of overlooked temporal clues. Secondly, we introduce the Motion-aware State Space Model (MSSM) to endow the model with the capacity to understand the temporal correlations of the same object across different time steps. Specifically, MSSM emphasizes the motion states of the same object at different time steps through two distinct temporal modeling and correlation steps. We utilize an improved state space model to represent these motion differences, significantly modeling the motion states. Finally, extensive experiments on the SemanticKITTI-MOS and KITTI-Road benchmarks demonstrate that the proposed MambaMOS achieves state-of-the-art performance. The source code of this work will be made publicly available.

## CCS CONCEPTS

• **Computing methodologies** → *Artificial intelligence*; *Vision for robotics*;

## KEYWORDS

Moving Object Segmentation; State Space Model; Spatio-Temporal Fusion

## 1 INTRODUCTION

LiDAR-based Moving Object Segmentation (MOS) task is pivotal for accurately delineating moving entities such as cars or pedestrians within the current LiDAR scan, serving as a fundamental component of autonomous perception [5, 46]. MOS contributes in two main ways. First, it ensures stable operation for autonomous driving systems by providing accurate 3D dynamic semantic scene understanding [8, 39]; Second, it assists in removing the "ghost

Permission to make digital or hard copies of all or part of this work for personal or classroom use is granted without fee provided that copies are not made or distributed for profit or commercial advantage and that copies bear this notice and the full citation on the first page. Copyrights for components of this work owned by others than the author(s) must be honored. Abstracting with credit is permitted. To copy otherwise, or republish, to post on servers or to redistribute to lists, requires prior specific permission and/or a fee. Request permissions from permissions@acm.org.
MM '24, October 28 − November 01, 2024, Melbourne, Australia
© 2024 Copyright held by the owner/author(s). Publication rights licensed to ACM.
ACM ISBN 978-1-4503-XXXX-X/18/06
https://doi.org/XXXXXXX.XXXXXXX

effect" caused by object motion during mapping in simultaneous localization and mapping, resulting in a clean static map [7, 24].

Chen *et al.* [5] propose a learning-based MOS method that projects point clouds onto a planar representation and utilizes a sequence of these representations to incorporate temporal information for MOS. Similar paradigms like [8, 20, 32, 39, 46] achieve low latency but suffer from geometric loss introduced by projection, leaving room for improvement in terms of accuracy and generalization. Non-projection methods [29, 41] perform feature extraction directly in the 3D space and have achieved precise segmentation results and excellent generalization. However, these methods cannot sufficiently couple the temporal-spatial features of multi-scan point clouds and suffer from the issue of "weak coupling between temporal and spatial information". Specifically, due to the changing spatial positions of moving objects over time, trailing artifacts will be formed in the aggregated point cloud. Without incorporating timestamp information to differentiate each scan in the aggregated point cloud, these artifacts may be confused with larger objects in terms of their similar appearance (*e.g.*, moving cars and parked trucks). The evolution of timestamp information reflects the motion of objects, and the moving objects can also be identified through the evolution of their timestamp information.

Based on the above observations, we hypothesize that the temporal information of objects is the dominant information for determining their motion, and strengthening the coupling between the temporal and spatial information of objects will facilitate the segmentation of moving objects. However, the aforementioned methods [29, 41] directly concatenate the timestamp information of each point with the spatially occupied information to form a 4D point cloud that contains temporal-spatial features and employ a Convolutional Neural Network (CNN) to learn these temporal-spatial features, as shown in Figure 1 (a). Although they are effective, the neglect of the dominant role of timestamp information and the lack of deeper coupling between temporal and spatial information hinder further improvement in their segmentation performance.

In this work, we rethink the issue of effectively encoding shallow temporal and spatial features and facilitating sufficient interaction among deep temporal and spatial features. For cases where simply concatenating timestamp information with spatial information fails to highlight the importance of temporal information, we propose an effective embedding approach named Time Clue Bootstrapping Embedding (TCBE), which emphasizes the expressive power of temporal information through attention mechanisms and enhances the mutual coupling between temporal and spatial information by treating temporal information as an independent channel separate from spatial information.

Although TCBE can enhance the coupling between temporal and spatial information to some extent compared to the previous embedding approach, it can only be applied to shallow layers and cannot further deepen the coupling between temporal and spatial

**Figure 1: A brief comparison of other non-projection methods (sub-figure (a)) with ours (sub-figure (b)). The previous methods treated temporal information $t$ and spatially occupied information $O$ equally, without deeply integrating them. In contrast, our method emphasizes the primacy of temporal information at each point more through our designed TCBE and achieves a deeper coupling of temporal and spatial information with MSSM, which aligns more closely with the fundamental principles of motion recognition.**

information. Recently, the work of [23] introduced a projection-based method and for the first time incorporated the self-attention mechanism core of transformers [40] into MOS, achieving better performance far surpassing the methods of the same paradigm. However, studies [10, 15] have shown that the quadratic computational complexity of the transformer model in managing large input sequences presents challenges in achieving a balance between training cost and accuracy. Fortunately, the State Space Model (SSM) introduced by Mamba [15] offers a promising solution, providing us with the opportunity to achieve comparable long-range context modeling capabilities to the transformer [40] while maintaining linear time complexity. Inspired by this advancement, we proceed to develop the Motion-aware Space State Model (MSSM). In the designed MSSM, we decouple the aggregated point cloud features into multiple single-scan features and learn the appearance features expressed by single-scan features and the motion features expressed by the aggregated features separately. Then, by using the cross-product attention between these two features, we achieve spatial appearance interpretation of multiple-scan features from single-scan features and temporal information supplementation of single-scan features from multiple-scan features, thereby enabling the deep-level coupling between temporal and spatial information with the assistance of SSM and achieving linear complexity.

Through extensive experiments, we demonstrate that the combination of TCBE and MSSM can effectively achieve strong coupling

between temporal and spatial information, and achieve state-of-the-art performances on SemanticKITTI-MOS [3, 5] and KITTI-Road [13] benchmarks. Our contributions are summarized as follows:

- We rethink the problem of the weak coupling between temporal and spatial information that existed in the previous methods and propose a novel LiDAR-based moving object segmentation framework, here in MambaMOS. To the best of our knowledge, this work represents the first attempt to utilize SSM in MOS, providing directions for future extensions of SSM in the MOS domain.
- An effective Time Clue Bootstrapping Embedding method (TCBE) is introduced, which enhances the coupling capability of temporal and spatial information to some extent, improving the expressive power of motion object features.
- A novel temporal-spatial information coupling module based on SSM (MSSM) is proposed, which enables deep-level coupling between temporal and spatial features and enhances the perception of moving objects through the complementary nature of single-scan and multiple-scan features.

## 2 RELATED WORK

Existing MOS methods can be categorized into two categories: Projection-based methods [5, 8, 20, 39] and Non-Projection-based methods [23, 24, 29, 30, 41]. Projection-based methods involve projecting a 3D point cloud onto a compact 2D plane as the model input, while Non-Projection-based methods are processed directly within the 3D point cloud space.

### 2.1 Projection-based methods

Projection-based MOS methods can be divided into Range View (RV) methods [5, 8, 20, 39] and Bird's-Eye View (BEV) ones [32, 46]. There has been extensive work in the field of object detection and segmentation in 3D LiDAR data using RV images [9, 12, 21], which use the original single scan point cloud through the spherical projection [31] to obtain 2D RV image as the model input. In motion perception tasks, the temporal information needed to perceive motion is usually provided by the residual images obtained from the residual processing of the RV images of the current scan and the past few scans [5, 8, 20, 39]. Chen *et al.* [5] directly concatenate the RV images and the corresponding multi-scan residual images as input, whereas Sun *et al.* [39] proposed a dual-branch model structure, which used two encoders to extract features from the RV images and the multi-scan residual images respectively. Different from [39], Kim *et al.* [20] use a branch in its model to decouple the movable objects into moving objects and static objects using additional semantic labels, which enhances the model's capacity to understand the dynamic scenario. Cheng *et al.* [8] focus more on the feature extraction of motion features, which coincides harmoniously with our viewpoint and have achieved leading performance with additional semantic labels.

Unlike the above RV-based methods, the BEV methods present the point cloud features from a top-down perspective, which maintains the consistency of the object scale in the point cloud and makes it easier to understand and process features [45]. Mohapatra *et al.* [32] first proposed moving object segmentation in BEV,

which achieved faster running speed but lower accuracy than RV-based methods. Zhou et al. [46] employed polar coordinates to transform point clouds into Bird's-Eye View (BEV) representation. They utilized a dual-branch CNN to extract appearance and motion features from multiple BEV scans, resulting in improved accuracy and efficiency. Although the above projection-based methods are efficient, there is a loss of geometric information in the process of returning the final result to the 3D point cloud space, which limits the performance of such methods.

## 2.2 Non-projection-based methods

Non-projection-based methods, which directly operate on point clouds in 3D space, circumvent the loss of geometric information inherent in projection-based approaches. Consequently, these methods hold the theoretical advantage of achieving superior segmentation performance. 4DMOS [29] inputs the voxelized point cloud superposition representation of multiple scans into a sparse 4D CNN and fuses the prediction results of multiple different scans of moving objects by a binary Bayesian filter as an additional post-process, which improved the confidence score of judging the moving object in the current scan and achieved excellent segmentation results. Similarly, InsMOS [41] is also based on a 4D point cloud as input, but they assist in segmenting moving objects by fusing BEV representations containing object instance information at different resolutions. Li et al. [23] proposed a dual-branch model that integrates 3D point clouds and 2D images, and employed Transformer [40] to fuse multi-scale point cloud and image features, aiming to enhance the coupling of temporal and spatial characteristics. Li et al. [24] utilized cylindrical coordinates to voxelize the aggregated point cloud input and employed a CNN to obtain moving object segmentation results and further applied MOS in the task of LiDAR-based localization to improve its robustness in dynamic scenes. MapMOS [30] improves that selecting fixed past scans will lead to some moving objects not being perceived due to occlusion. Therefore, a strategy of moving target perception based on a local map constructed by past scans is proposed, and state-of-the-art performance is achieved on the validation set of the SemanticKITTI-MOS benchmark. In addition to the learning-based methods mentioned above, there are also many non-learning-based methods, including map-cleaning methods [2, 19, 25, 26, 36] and map-based methods [6, 34, 37]. Map-cleaning methods remove the moving objects offline by the geometric information of the target [2, 19, 25, 26, 36]. Map-based methods, on the other hand, require a pre-built map to remove the objects that are moving throughout the mapping process [6, 34, 37].

In general, existing MOS methods have not thoroughly explored the coupling between temporal and spatial features, which limits their understanding of motion states. In contrast, our method achieves shallow coupling of temporal and spatial features during the embedding stage and deep coupling within each stage of the model. This deep coupling establishes a robust correlation between temporal and spatial clues, enhancing the model's comprehension of motion scenes. Importantly, our method achieves state-of-the-art performance without any post-processing modules on the MOS task.

## 3 METHOD

### 3.1 Preliminaries

**State Space Model.** SSM [14] is a sequential model that can map a one-dimensional input $x(t) \in \mathbb{R}$ sequence to an output sequence $y(t) \in \mathbb{R}$. The process is represented by a series of continuous hidden states $h(t) \in \mathbb{R}^N$ of state size $N$. In general, the SSM of a continuous-time system can be represented by the following linear Ordinary Differential Equation (ODE) as depicted in Equation (1),

$$h'(t) = Ah(t) + Bx(t)$$
$$y(t) = Ch(t) \tag{1}$$

where the parameters $A \in \mathbb{R}^{N \times N}$, $B \in \mathbb{R}^{N \times 1}$ and $C \in \mathbb{R}^{1 \times N}$ establish the correlation between the state and output variables.

**Discretization.** It is essential that the original SSM equations be transformed into a discrete form to fit the discretized data in the task. The discretized SSM can be written as Equation (2),

$$h_t = \bar{A}h_{t-1} + \bar{B}x_t$$
$$y_t = Ch_t \tag{2}$$

The discretization parameters $\bar{A}$, $\bar{B}$ can be described by the Zero-Order Hold (ZOH) rule with timescale parameters $\Delta$ as Equation (3),

$$\bar{A} = e^{\Delta A}$$
$$\bar{B} = (\Delta A)^{-1} \left( e^{\Delta A} - I \right) \cdot \Delta B \tag{3}$$

**Selective Scans Mechanism.** Mamba [15] proposed a selective scan mechanism that effectively adjusts the parameters through the parameterized projection of the input sequence, enabling the SSM to selectively filter the input sequence features. This has advanced the research of SSM in the time-varying domain.

**MambaMOS Principles.** To address the weak coupling of temporal and spatial information in existing MOS methods, we try to adapt Mamba from a Natural Language Processing (NLP) to MOS task. An intriguing discovery emerged: the MOS task inherently involves selecting a moving subset of elements from an unordered set, akin to the selective copying mechanism in NLP [1, 15]. Leveraging this insight, we introduced MambaMOS based on the selective copying mechanism. This enhancement equips Mamba to effectively address MOS tasks, enabling the model to adaptively select the moving target while reducing operational costs.

### 3.2 MambaMOS

**Overview Architecture.** The proposed MambaMOS leverages a U-Net [35] style overall architecture as shown in Figure 2. Firstly, the 4D point cloud set as input will be transformed into an ordered sequence after the serialization process.

Simultaneously, they are encoded through the meticulously designed TCBE (Sec. 3.3). Next, the point cloud is sent into the encoder-decoder construct to model deep features. It includes the encoder with a 5-stage block depth of [2, 2, 2, 6, 2] and the decoder with a 4-stage block depth of [2, 2, 2, 2]. It should be noted that the point cloud pooling strategy is used in the encoder of all stages except for the first. The scale change factor of the point cloud passing the pooling layer is 2. Moreover, at the beginning of the block, an efficient position encoding block is leveraged to capture the local

Figure 2: The overview of our proposed MambaMOS. The previous $F − 1$ scans, after undergoing viewpoint transformation, are overlaid with the current scan to form a 4D point cloud. This 4D point cloud is then serialized to obtain a sequence as input. After passing through TCBE, the coupling degree between temporal and spatial information in the input is enhanced and fed into a symmetric encoder-decoder architecture (the pink box). Each stage of the encoder/decoder consists of a pooling/unpooling layer and $N$ blocks (the blue box). MSSM serves as the core of each block to achieve deep-level coupling of temporal and spatial features. Finally, the MOS result in the current scan can be obtained from the output of the decoder by a linear layer.

attention of the feature following the idea of most point transformer works [22, 43, 44].

The point cloud features after layer normalization will pass through the MSSM (Sec. 3.4), the core insight of the entire block, where the motion features of the objects will be enhanced. The final output of the block is the layer normalization and a multi-layer perceptron. And residual connections are extensively applied in each of our blocks to avoid vanishing gradients [16]. Finally, the logits of each point can be obtained by a linear layer. And points are deserialized to extract the segmentation result.

**Input Representation.** At the current time ($t = 0$), given a LiDAR scan $S_t = \{p_i \in \mathbb{R}^4\}_{i=0}^{N_t-1}$ with $N_t$ points $p_i = [x_i, y_i, z_i, 1]^T$ represented by homogeneous coordinates. The goal is to segment the moving points in the current scan $S_0$ using the continuous point cloud set $S = \{S_t\}_{t=0}^{F-1}$ including the current scan $S_0$ and its past $F − 1$ scans. To aggregate the $F$ scans point cloud data into a 4D point cloud input containing temporal-spatial information and eliminate self-motion, we need to transform the past $F − 1$ scans to the perspective of the current scan and convert the homogeneous coordinates to Cartesian coordinates separately. Given the pose transition matrix $T_t^0$ from scan $t$ to the current scan, the perspective

transition from the point cloud at time $t$ to the current point cloud can be expressed as Equation (4).

$$S_{t\to0} = \left\{p_i' = T_t^0 \cdot p_i \mid p_i \in S_t\right\}_{i=0}^{N_t-1} \quad (4)$$

Thus, the 4D point cloud set $S' = \left\{p_i' \in \mathbb{R}^4\right\}_{i=0}^{N-1}$ with $N = \sum_{t=0}^{F-1} N_t$ points can be represented as Equation (5). To distinguish each scan in the 4D point cloud, we add the corresponding time step of each scan as an additional dimension of the point and obtain the spatio-temporal point representation $p_i' = [x_i, y_i, z_i, t_i]^T$.

$$S' = \{S_0, S_{1\to0}, ..., S_{t\to0}\} \quad (5)$$

**Serialization.** The SSM, as the core of MambaMOS, typically takes in a sequence of data, such as natural language. Therefore, it is necessary to obtain the sequence $S_o'$ from the unordered 4D point cloud set $S'$ by serialization. The serialization can be understood as a projection function $\Psi$ that transforms the unordered set $S'$ into the sequence $S_o'$. Thus, the process of serialization and deserialization can be described as Equation 6, where $\Psi^{-1}$ is the inverse projection function. One approach to serialize point clouds is by sorting the coordinate of each point [27]. However, this serialization method fails to adequately preserve the local spatial relationships of the

objects, which may result in spatially close point clouds being far apart in the final sequence.

Space-filling curves are mathematical curves that can project data in $N$-dimensional space to one-dimensional continuous space: $\mathbb{Z}^N \rightarrow \mathbb{Z}$, which have been applied in recent 3D scene understanding works [42, 43]. Inspired by them, our serialization process utilizes z-order curves [33] and Hilbert curves [17], which preserve neighborhood relationships in the original 3D point cloud effectively.

$$S'_o = \Psi(S')$$
$$S' = \Psi^{-1}(S'_o) \tag{6}$$

## 3.3 Time Clue Bootstrapping Embedding

Previous methods [29, 30, 41] have not effectively emphasized the dominance of temporal information for each point. This is evident in their equal treatment of spatial occupied information obtained from LiDAR and the corresponding timestamp information from the scan aggregation process. However, the direct overlaying for temporal and spatial information, which belong to different modalities, does not fully exploit the supervisory role of one modality on the other. Therefore, we propose the Time Clue Bootstrapping Embedding (TCBE). It emphasizes temporal information over spatial information based on the principle that time evolution drives the motion of objects, thereby enhancing the coupling of temporal and spatial information.

The structure of TCBE is illustrated in the bottom right of Figure 2. Specifically, TCBE embeds the spatial and temporal information of each point in the ordered point cloud sequence $S'_o$ using 1D convolution to obtain the corresponding spatial feature $f_s$ and temporal feature $f_t$ in embedded dimension, both of which possess local characteristics. Firstly, the initial coupled temporal and spatial feature $f_{cou}$ is obtained by adding the temporal feature $f_t$ and the spatial feature $f_s$, which serves as an alternative implementation of previous embedding methods. Then, in order to emphasize the dominance of temporal information over spatial information, $f'_t$ which reflects the local temporal evolution trends, obtained through 1D convolution without changing its channels, is multiplied element-wise by $f_s$. Finally, the time-guided spatial feature $f_{TGS}$ is added to the initial coupled temporal and spatial feature $f_{cou}$, resulting in the enhanced temporal and spatial coupling information $f'_{cou}$. After undergoing 1D convolution, batch normalization, and activation functions, $f'_{cou}$ serves as the output of TCBE.

## 3.4 Motion-aware State Space Model

Although there are some similarities in form between the selective copy task [1, 15] in NLP and the MOS task as mentioned before, the direct application of Mamba [15] cannot effectively exploit the temporal features. This is attributed to the fact that the original Mamba [15] is designed for one-dimensional natural language with a certain causal relationship. However, the serialized multi-scan point cloud sequence cannot reflect strong causality. Thus, we propose MSSM to compensate for the shortcomings of Mamba [15] on MOS.

The main design idea of MSSM is to enhance the original Mamba's perception of temporal features regarding moving objects by using cross-product attention between single-scan features and multi-scan features. As shown in the upper left of Figure 2, it is mainly composed of linear layers, activation function $\sigma$, and an SSM with the selective scans mechanism. Let the input point cloud feature with batch size $B$, sequence length $N$, and number of channels $C$ be characterized by $f_I \in \mathbb{R}^{B \times N \times C}$, which will go through three branches. We derive the main branch of Mamba [15] to obtain the upper and the middle branches of our MambaMOS. The upper branch is used to extract the appearance features of each object in the single-scan point cloud. And the middle branch focuses more on the temporal features of moving objects in the 4D point cloud. Since the MOS task only focuses on moving objects, we aim for the MSSM to assign lower attention to unmovable objects such as roads or tree trunks. Therefore, a feature weighting process is required. Inspired by the gated attention units [18], we employ a simple gating mechanism as the bottom branch of MSSM to allocate weights to features in each hidden state, thereby determining whether the features are expressed.

Specifically, to obtain the single scan feature $f_S \in \mathbb{R}^{B' \times N_p \times C}$ with $B' = B \times F$ at this time, the upper branch firstly performs Reversed Aggregation (RA), which separates each scan of $S'$ and concatenates them as a separate batch after 0-padding to $N_p$. Then the appearance features of the single scan $f'_A$ can be obtained through the process of 1D convolution and single scan aggregation. This process can be written as:

$$f'_A = \sigma\left(\text{Conv1d}\left(\text{RA}\left(f_I\right)\right)\right) \tag{7}$$

The middle branch employs 1D convolution to obtain the temporal and appearance features of moving objects in multiple scans. The output of this process is denoted as $f'_M$. Subsequently, $f'_M$ is fused with the output of upper branch $f'_A$ through the cross-product attention to obtain $f_{MG}$. The fusion process can be described as follows:

$$f_{MG} = \text{Sigmoid}\left(f'_M\right) \otimes f'_A + f'_M \tag{8}$$

In the subsequent design, we follow the idea of the original Mamba, that is, the final output $f'$ of the block is obtained by element-wise multiplication of the result of the main branch and the result $f_G$ of the gated branch after a linear projection. This process is described as follows:

$$f' = \text{SSM}\left(\sigma\left(f_{MG}\right)\right) \otimes f_G \tag{9}$$

## 3.5 Loss Function

Before performing the loss calculation, we first deserialize the obtained sequence segmentation results to correspond to the initial unordered point cloud set as Euqation 6. Afterwards, following the majority of 3D segmentation methods, we adopt the combination of cross-entropy loss ($\mathcal{L}_{ce}$) and Lovász-Softmax loss [4] ($\mathcal{L}_{ls}$) as the joint loss $\mathcal{L} = \mathcal{L}_{ce} + \mathcal{L}_{ls}$ for supervised training.

## 4 EXPERIMENT

### 4.1 Experiment Setups

We perform a variety of experiments to verify the proposed MambaMOS on the SemanticKITTI-MOS dataset [3, 5]. Sequences 00~07 and 09~10 are used as the training set, sequence 08 is used as the

 

validation set, and sequences 11~21 are used as the test set, following the same division as previous MOS methods [8, 30, 41]. The KITTI-road dataset [13] is also utilized in the experiments for comparison with other MOS methods, and the same partitioning approach as [8] is maintained. The entire training is conducted on four NVIDIA RTX A6000 GPUs with 48G VRAM for 50 epochs with a batch size of 4. AdamW [28] with a weight decay of 0.005 is used as the optimizer, and the learning rate is set to 0.00032. A grid size of $0.09m$ is applied to voxelize the input aggregated point cloud, and scans of $F = 8$ are used for input same as [5, 8, 39]. Moreover, common point cloud data augmentation approaches such as random rotation and random flipping are applied during training to enhance the generalization capacity of MambaMOS. All ablation experiments were conducted on eight NVIDIA GeForce RTX 3090 GPUs, using a four-scan input ($F = 4$), a batch size of 8, and completed using automatic mixed precision. We report the voxelized moving object IoU for ablations. Additionally, similar to [5, 8, 20], we employed additional semantic labels for training as well. During the validation and testing stages, Intersection-over-Union (IoU) [11] is adopted as the metric to evaluate the performance. Following previous methods [23, 24, 30], all experiments provide IoU for the moving objects as $IoU_{MOS}$ as the main evaluation metric which can be described as Equation 10 with True Positive $TP$, False Positive $FP$ and False Negative $FN$:

$$IoU_{MOS} = \frac{TP}{TP + FP + FN} \qquad (10)$$

## 4.2 Moving Object Segmentation Performance

We analyze the comparison with other SoTA methods from both quantitative and qualitative perspectives.

**Quantitative Analysis.** Table 1 presents the quantitative comparison results of MambaMOS with state-of-the-art methods in the MOS benchmark [3, 5]. All results reported for each method are the best-reported results in their respective papers. Due to some methods [8, 39, 41] using the KITTI-Road dataset [13] as additional training data in their comparisons, we follow the principle of fair comparison by using consistent training data, distinguishing it with a symbol † from the original comparison on the table.

Without using additional training data, MambaMOS successfully outperforms almost all methods on the benchmark. Specifically, MambaMOS surpasses Two-streamMOS [23], which integrates point clouds and images as input, by 4.4% and 2.6% on the validation set and hidden test set. We attribute this significant improvement largely to the geometric losses present in non-projection-based methods. In the comparison with non-projection-based methods, MambaMOS achieves superior performance on the validation than LiDAR-IMU-GNSS [24], which includes ground optimization as an additional pre-processing, by 3.3%, and leads it by 0.7% on the test set, owing to the stronger coupling of temporal and spatial information in MambaMOS. Despite MambaMOS using a fixed number of scans as input, it still surpasses MapMOS [30], which leverages a local map instead, with a significant margin of 9.6% on the hidden test set. After incorporating additional training data, MambaMOS, when trained with the same settings as before, still outperforms other methods under comparable conditions. MambaMOS surpasses the state-of-the-art MF-MOS [8] by 3.4%, and outperforms InsMOS [41],

**Table 1: Comparison with state-of-the-art methods on the SemanticKITTI-MOS benchmark. † denotes using additional KITTI-Road for training.**

| Method | $IoU_{MOS}$ (%) | |
|---|---|---|
| | Validation 08 | Test 11-21 |
| LiMoSeg [32] | 52.6 | - |
| LMNet [5] | 67.1 | 54.5 |
| SSF-MOS [38] | 70.1 | - |
| MotionSeg3D [39] | 71.4 | 64.9 |
| RVMOS [20] | 71.2 | 74.7 |
| 4DMOS [29] | 77.2 | 65.2 |
| InsMOS [41] | 73.2 | - |
| MF-MOS [8] | 76.1 | - |
| MotionBEV [46] | 76.5 | 69.7 |
| MapMOS [30] | **86.1** | 66.0 |
| Two-streamMOS [23] | 77.9 | 73.0 |
| LiDAR-IMU-GNSS [24] | 79.0 | 74.9 |
| MambaMOS | 82.3 | **75.6** |
| LMNet† [5] | 63.8 | 60.5 |
| MotionSeg3D† [39] | 69.3 | 70.2 |
| MotionBEV† [46] | 64.6 | 74.9 |
| InsMOS† [41] | 69.4 | 75.6 |
| MF-MOS† [8] | - | 76.7 |
| MambaMOS† | **73.3** | **80.1** |

which utilizes an additional instance bounding box for determining moving instance, by 3.9% and 4.5%, on the validation set and hidden test set respectively.

To further analyze the advantages brought by our method, we have conducted a detailed comparison of the segmentation performance of existing methods on the SemanticKITTI-MOS validation set for different distances, as shown in Table 2. The metrics in Table 2 are either reported in their respective papers or determined using their publicly available weights. The weights for other methods such as RVMOS [20], Two-streamMOS [23], and LiDAR-IMU-GNSS [24] are either undisclosed or not reported in the papers, hence not included in the comparison.

As known, the point cloud distribution becomes sparser as the object distance from the LiDAR increases. As shown in Table 2, most MOS methods achieve satisfactory segmentation results at close distances. However, their segmentation performance sharply declines when the distance reaches the range of $20m$ and $50m$. Furthermore, beyond a distance of $50m$, some projection-based methods such as LMNet [5], MotionSeg3D [39], and MotionBEV [46] fail to discern the motion attributes of the objects. Although MF-MOS [8], with its focus on motion features, surpasses the non-projection-based methods like 4DMOS [29] and InsMOS [41] in segmenting distant moving objects, it is still limited in recognizing the motion attributes of distant objects due to geometric losses caused by the projection process, which prevents the strong coupling of spatial information and temporal information for the objects. On the other hand, MambaMOS demonstrates precise segmentation of moving objects even in cases of extremely sparse point clouds. This indirectly supports

**Table 2: MOS performance on the SemanticKITTI-MOS validation set for points at different distances. R denotes recall and P denotes precision.**

| Method | Close (<20m) | | | Medium (>=20m, < 50m) | | | Far (>= 50m) | | |
|---|---|---|---|---|---|---|---|---|---|
| | $IoU_{MOS}$ | R | P | $IoU_{MOS}$ | R | P | $IoU_{MOS}$ | R | P |
| LMNet [5] | 70.72 | 76.89 | 89.80 | 43.88 | 54.30 | 69.56 | 0.00 | 0.00 | - |
| MotionSeg3D$^†$ [39] | 71.66 | 79.97 | 87.35 | 52.21 | 59.27 | 81.40 | 4.99 | 4.99 | **100.00** |
| MotionBEV [46] | _80.85_ | 85.40 | 93.81 | 56.35 | 59.89 | 90.50 | 0.00 | 0.00 | - |
| 4DMOS [29] | 78.43 | 82.11 | _94.59_ | _68.71_ | _72.62_ | **92.74** | 41.00 | 41.00 | **100.00** |
| InsMOS [41] | 75.29 | **88.78** | 83.21 | 57.67 | 66.81 | 80.84 | 10.88 | 10.89 | _98.63_ |
| MF-MOS [8] | 79.31 | 84.98 | 92.23 | 54.67 | 64.10 | 78.81 | _47.97_ | _50.08_ | 91.94 |
| MambaMOS | **83.69** | _87.30_ | **95.29** | **72.48** | **78.24** | _90.78_ | **94.44** | **97.73** | 96.56 |

the viewpoint delivered in our work: reinforcing temporal information can effectively improve the MOS performance when the spatial features of the targets are not prominent.

**Qualitative Analysis.** As shown in Figure 3, MambaMOS has achieved significant performance improvements compared to other methods on the SemanticKITTI-MOS validation set [3, 5]. We have conducted a detailed analysis of this phenomenon as shown in the figure. MF-MOS [8], InsMOS [41], and 4DMOS [29] all fail to correctly determine the motion attributes of the objects, resulting in a large number of false negative results. It can be attributed to they only rely on a weak coupling between spatial and temporal information for motion estimation. Slow-moving objects or distant moving objects do not exhibit obvious characteristics in terms of spatial information, so methods [8, 29, 41] that do not strengthen the temporal information perform poorly in their estimation. However, MambaMOS effectively addresses this problem by incorporating strong temporal information coupling. Furthermore, to reduce false positive predictions, we also categorize stationary vehicles as a movable class for training, following the method of [5, 8, 20], which further enhances the model's understanding of motion scenes.

## 4.3 Ablation Study

Since the SSM receives sequential features, different spatial serialization combinations will have an impact on overall performance. We explore the influence of the serialization combination of $z$-curve [33] and Hilbert curve [17], which have good spatial locality characteristics, on the performance of MOS. As the spatial filling curves traverse the spatial points based on the order of $x$, $y$, and $z$, prioritizing $y$ will yield different serialization results compared to prioritizing $x$. We denote this variant with $^T$. As shown in Table 6, richer serialization methods yield better performance on the validation set. This is because multiple serialization methods capture different contextual relationships of sequences, reducing overfitting while enhancing the model's understanding of dynamic objects.

Table 4 presents two methods proposed in our study to enhance the coupling of temporal and spatial information. We conducted ablation experiments to demonstrate that the proposed modules can enhance the model's perception of moving objects. When MSSM is not used, we replaced it with the original Mamba block [15], and we employed a simple 3D convolution for information embedding when TCBE is not applied.

**Table 3: Ablation about the serialization combination on the SemanticKITTI-MOS validation set.**

| Pattern | $IoU_{MOS}$ (%) |
|---|---|
| Z | 75.11 |
| Hilbert | 76.38 |
| Z+Z$^T$ | 76.57 |
| Hilbert+Hilbert$^T$ | 76.41 |
| Z+Z$^T$+Hilbert+Hilbert$^T$ | **77.46** |

**Table 4: Ablation about each module in MambaMOS on the SemanticKITTI-MOS validation set.**

| Component | | $IoU_{MOS}$ (%) |
|---|---|---|
| MSSM | TCBE | |
| ✗ | ✗ | 75.21 |
| ✓ | ✗ | 77.07 |
| ✗ | ✓ | 76.62 |
| ✓ | ✓ | **77.46** |

From the experimental results, it can be observed that when only MSSM or TCBE is applied compared to the baseline, the performance is improved by 1.86% and 1.41%. This indicates that both MSSM and TCBE can enhance the coupling of temporal and spatial features and improve the model's perception of motion features. However, when only MSSM is applied, the performance is improved by 0.45% compared to TCBE. This is because MSSM, based on the interaction between single-scan features and multi-scan features, focuses more on deep-level spatio-temporal information coupling and can learn the motion attributes of objects more comprehensively. Finally, by joining TCBE on the basis of MSSM, the emphasis on temporal information during the embedding phase is further enhanced, which aligns with the fundamental logic of motion recognition and achieves optimal performance, surpassing the baseline by 2.25%.

## 4.4 Generalization Performance Analysis

Since the majority of the SemanticKITTI dataset [3] was collected in residential areas, to test the broader environmental adaptability

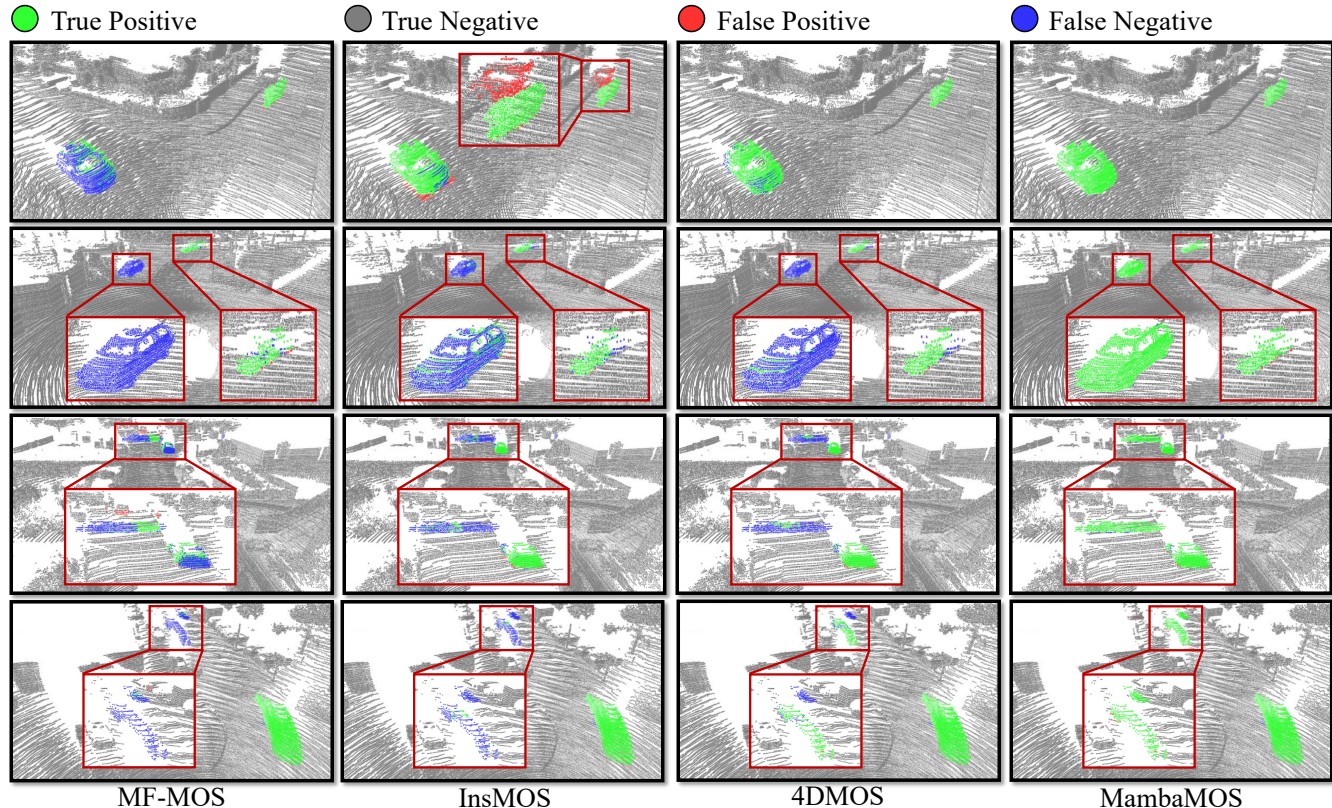

**Figure 3: Visualization comparison results of MambaMOS with MF-MOS [8], InsMOS [41], and 4DMOS [29] on the SemanticKITTI validation set. We overlay their respective predictions for the current scan and the past seven scans to visually demonstrate the results of MOS.**

of MambaMOS, we fine-tune it on the KITTI-Road dataset [13] to evaluate its generalizability to new environments. We follow the same data partitioning strategy as MF-MOS [8], InsMOS [41], and MotionSeg3D [39], and compared it with methods that use fixed scans as input with publicly available weights. The original weights of all methods shown in Table 5 are open-source and trained exclusively on the SemanticKITTI-MOS [3, 5] dataset. They were then fine-tuned for 10 epochs on the KITTI-Road training set [13] to obtain the final results. The results indicate that even with a small amount of data and minimal fine-tuning, MambaMOS still achieves better results than previous methods, demonstrating its excellent generalization capability for adapting to new environments.

## 5 CONCLUSION

This paper introduces MambaMOS, a novel framework for moving object segmentation, aiming to address the issue of weak spatio-temporal coupling in existing methods. Specifically, we introduce the Time Clue Bootstrapping Embedding to achieve the shallow coupling of temporal and spatial information of the objects. Furthermore, we underscore the importance of temporal information as the primary cue for recognizing motion attributes, thereby enhancing the model's sensitivity to motion features. To achieve deeper spatio-temporal coupling, we propose the Motion-aware State Space

**Table 5: Comparison of fine-tune performance with state-of-the-art methods on the KITTI-Road dataset.**

| Method | $\text{IoU}_{MOS}$ |
|---|---|
| LMNet [5] | 87.4 |
| MotionBEV [46] | 80.5 |
| 4DMOS [29] | 81.0 |
| InsMOS [41] | 83.9 |
| MF-MOS [8] | 87.9 |
| **MambaMOS** | **89.4** |

Model, which facilitates interaction between single-scan and multi-scan features. Leveraging the SSM's linear complexity and strong contextual modeling capability, the MSSM achieves strong spatio-temporal coupling of features. Extensive experiments validate the effectiveness of our method, demonstrating state-of-the-art performance on both SemanticKITTI-MOS and KITTI-Road datasets. Additionally, this paper marks the pioneering application of SSM to the MOS task, and establishes a significant connection between point cloud segmentation in 3D vision and natural language tasks, offering valuable insights for future research directions.

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
