# OpenReview forum: "MambaMOS: LiDAR-based 3D Moving Object Segmentation with Motion-aware State Space Model"
_acmmm.org/ACMMM/2024/Conference — MM2024 Poster_

### Official Review · Reviewer_nwPp · 2024-05-19

**Rating:** 4
**Confidence:** 4

**Summary:**

In summary, thish paper discussed the challenges of learning-based point cloud MOS and proposed a method for moving object segmentation, which utilize an improved state space model to represent these motion differences, modeling the motion states.

**Strengths:**

- Excellent performance performance, basically exceeds the existing solution.
- The paper is well-written and presents a novel contribution to the field of point cloud MOS.

**Limitations:**

There are some key points and suggestions that the authors should address:

- L342, what do [2,2,2,6,2] and [2,2,2,2] represent respectively? What is 6?
- I would like to confirm with the author whether the input of the point cloud only contains the xyz coordinate position and not the intensity information of the point cloud?
- Looking at Fig2, the loss function only supervises the current frame t? Or t-1, t-2 will be supervised at the same time?
- Why does the final experiment use 8 frames as input, while the ablation experiment uses 4 frames as input? Have you also done ablation experiments on the number of input frames? The more input frames the better? 4DMOS seems to input 10 frames, while LMNet, MotionSeg3D, inputs 8 frames, just as you said.
- This article does not give the inference time/speed of the proposed method, but this is compared and reported in 4DMOS LMNet and MotionSeg3D. It is recommended that the author provide a fair model inference speed for reference.
- The video of additional materials cannot show the comparison with other solutions, and the comparison with GT, which has almost no reference value.

**Suitability:**

2

---

### Official Review · Reviewer_mvRD · 2024-05-25

**Rating:** 4
**Confidence:** 3

**Summary:**

This paper aims to achieve strong coupling of temporal and spatial information in LiDAR sequences to improve the accuracy of 3D Moving Object Segmentation.
The main contributions include an Time Clue Bootstrapping Embedding method which enhances the shallow coupling of temporal and spatial information, a temporal-spatial information coupling module that enables deep-level coupling.
The proposed method is the first attempt to ultize SSM in MOS and achieves state-of-the-art performance on the SemanticKITTI-MOS and KITTI-Road benchmarks.

**Strengths:**

The proposed method utilizes the Time Clue Bootstrapping Embedding and temporal-spatial information coupling to enhance the shallow and deep coupling of temporal and spatial information, respectively.
It achieves state-of-the-art performance on the SemanticKITTI-MOS and KITTI-Road datasets.

**Limitations:**

1. Do not use the same numbering for multiple formulas, such as lines 300-302, lines 308-310, etc.
2. The serialization process in Section 3.2 is unclear. The space-filling curves project the data in 𝑁-dimensional space to one-dimensional continuous space. In this paper, the z-order curves and Hilbert curves project the 4D point cloud to 1D? In Figure 2, it can be seen that the results are a two-dimensional point cloud with size of nx5. In addition, the Equation (6) is too concise to reflect the point cloud conversions.
3. Compared to the existing moving object segmentation methods, what is the time cost of the proposed method?

**Suitability:**

2

---

### Official Review · Reviewer_TB9S · 2024-05-26

**Rating:** 3
**Confidence:** 4

**Summary:**

This paper proposes a transformer-based framework to segment 3D moving objects among LiDAR point cloud frames by emphasizing temporal information and coupling spatial-temporal cues.

**Strengths:**

1. A LiDAR-based sequential framework is designed for the 3D moving object segmentation task.

**Limitations:**

1. The motivation and method demonstration for "Time Clue Bootstrapping Embedding" in Section 3.3 are confusing. Explaining why temporal information is more important than spatial information for 3D Moving Object Segmentation would help to better illustrate your novelty. Additionally, the method demonstrations of this module in Figure 2 and in text lines 49-508 need to be much clearer for the audience. Formula expressions are necessary for understanding.

2. The same issue exists for the "Motion-aware State Space Model" in Section 3.4: the method demonstration is hard to follow.

3. The advantage of the Mamba-based transformer is its comparable performance with less computational complexity. This feature is mentioned in line 153 of the text, but there are no experiments provided to demonstrate the efficiency of this work.

4. Other questions: I am curious about the differences between 3D moving object segmentation and 3D scene flow estimation/segmentation.

**Suitability:**

2

---

### Meta-Review · Area_Chair_tgAx · 2024-07-03

**Recommendation:** Accept (Poster)
**Confidence:** 4

**Metareview:**

the paper proposes a LiDAR-based 3D Moving Object Segmentation with Motion-aware State Space Model. All reviewers agree the proposed method improves performance.
The rebuttal addresses the key questions raised by the reviewers.